# Identification and Characterization of a Potential Probiotic, *Clostridium butyricum* G13, Isolated from the Intestine of the Mud Crab (*Scylla paramamosain*)

Huifen Liang,[a,b] Ngoc Tuan Tran,[a,b] Taoqiu Deng,[a,b] Jinkun Li,[a,b] Yifan Lei,[a,b] Mohammad Akibul Hasan Bakky,[a,b] Ming Zhang,[a,b] Rui Li,[a,b] Wenxuan Chen,[a,b] Yueling Zhang,[a,b] Xiuli Chen,[c] Shengkang Li[a,b]

aGuangdong Provincial Key Laboratory of Marine Biology, Shantou University, Shantou, China
bInstitute of Marine Sciences, Shantou University, Shantou, China
cGuangxi Academy of Fishery Sciences, Guangxi Key Laboratory of Aquatic Genetic Breeding and Healthy Aquaculture, Nanning, China

**ABSTRACT** The butyrate-producing bacterium *Clostridium butyricum* has been proven to be important in improving the growth and health benefits of aquatic animals. In this study, *C. butyricum* G13 was isolated for the first time from the gut of the mud crab (*Scylla paramamosain*). The results of this study showed that *C. butyricum* G13 could produce a high concentration of butyric acid and grow well in a wide range of pHs (4 to 9) and NaCl (1 to 2.5%) and bile salt (0.2 to 1.0%) concentrations. *In vitro* characterization revealed that *C. butyricum* G13 is a Gram-positive and gamma-hemolytic bacterium sensitive to most antibiotics and shows hydrophobicity and the capacity to degrade starch. *In vitro* fermentation using mud crab gut contents showed that *C. butyricum* G13 alone or in combination with galactooligosaccharides (GOS) and/or resistant starch (RS) significantly increased butyric acid production and beneficially affected the abundance and diversity of intestinal microbiota. In addition, *C. butyricum* G13 can improve the survival rate of mud crabs and effectively maintain the normal structure of gut morphology after infection with *Vibrio parahaemolyticus*. In conclusion, *C. butyricum* G13 can be considered a potential probiotic that improves the immune capacity and confers health benefits on mud crabs.

**IMPORTANCE** With the development of society, more and more aquatic animals are demanded. Intensification in the aquaculture scale is facing problems, such as disease outbreaks, eutrophication of water bodies, and misuse of antibiotics. Among these challenges, disease outbreak is the most important factor directly affecting aquaculture production. It is crucial to explore new approaches effective for the prevention and control of diseases. Probiotics have been widely used in aquaculture and have shown beneficial effects on the host. In this study, the butyrate-producing bacterium *Clostridium butyricum* G13 was isolated for the first time from the intestine of the mud crab through *in vitro* fermentation. The bacterium has probiotic properties and changes the gut microbiota to be beneficial to hosts *in vitro* as well as protecting hosts from *Vibrio parahaemolyticus* infection *in vivo*. The outcomes of this study indicate that *C. butyricum* G13 can be used as a potential probiotic in mud crab aquaculture.

**KEYWORDS** mud crab, *Clostridium butyricum* G13, probiotic, gut microbiota, survival rate

Address correspondence to Shengkang Li, lisk@stu.edu.cn, or Ngoc Tuan Tran, tranntts@gmail.com.

The authors declare no conflict of interest.

[This article was published on 31 July 2023 with Taoqiu Deng's name misspelled as "Tiaoqiu" in the byline. The byline was updated in the current version, posted on 17 August 2023.]

The mud crab (*Scylla paramamosain*) is one of the important commercial aquaculture species in the southeast coastal area of China (1–4). The continuous expansion of the culture scale results in an increase in the frequency of disease outbreaks in farmed mud crabs, which leads to a gradual decline in aquaculture production and a great economic loss (5, 6). The infectious diseases that occurred in mud crabs were

mainly caused by members of the genus *Vibrio*, such as *Vibrio parahaemolyticus*, *V. cholerae*, and *V. vulnificus* (5, 7–10). Previous studies have reported the application of antibiotics (11, 12), vaccines (13, 14), and feed additives (including probiotics, prebiotics, and synbiotics) (15–17) used for prevention or control of diseases in aquatic animals. Among these methods, antibiotics are the most commonly used, showing high effectiveness in treating infectious diseases in cultured animals. However, the widespread use of antibiotics leads to direct impacts on farmed animals and environments as well as indirect effects on human health (18, 19). The gut microbiota plays an important role in the host's healthy growth (20). Interestingly, the use of probiotics has been discussed as one of the most suitable approaches to reducing the risk of infection in mud crabs (21). A previous study showed that dietary supplementation of *Bacillus subtilis* DCU or *Bacillus pumilus* BP improved the survival rate, enhanced the immune function, and reduced the infection rate of hosts after challenge with *V. parahaemolyticus* (2). In another study, Yang et al. (3) revealed that the bacteria *Enterococcus faecalis* Y17 and *Pediococcus pentosaceus* G11 could improve the growth performance and immunity of mud crabs. The results of these studies indicate the potential use of probiotics in mud crab aquaculture. Although the health benefits of probiotics for mud crabs have been displayed, the number of species used as probiotics is still limited, which includes only members of *Bacillus*, *Lactobacillus*, *Enterococcus*, and *Pediococcus* (21). Discovering new species for use as probiotic microorganisms is required to enrich the supply for mud crab farming.

*Clostridium butyricum* is an obligate anaerobic Gram-positive bacterium that produces large amounts of gas in medium containing fermentable carbohydrates (22, 23). *C. butyricum* can also ferment carbohydrates to produce short-chain fatty acids (SCFAs), especially butyric acid, which are used for the regeneration and repair of the intestinal epithelium and regulation of intestinal health microecosystems (24–26). The bacterium has been characterized as more resistant to low-pH, high-temperature environments and multiple antibiotics than other probiotics (such as *Bacillus*, *Lactobacillus*, and yeasts) and used as a probiotic in aquaculture (22, 27). Dietary supplementation of *C. butyricum* showed an important role in improving the growth performance and health benefits of several aquatic animals, such as spotted sand bass (*Paralabrax maculatofasciatus*) (28), Pacific white shrimp (*Litopenaeus vannamei*) (29, 30), black tiger shrimp (*Penaeus monodon*) (31), and grass turtles (*Chinemys reevesii*) (32), as well as crucian carp (*Carassius vulgaris*) intestinal epithelial cells (FIECs) (33). For instance, the addition of *C. butyricum* to the diet of *L. vannamei* and feeding for 42 days could effectively improve the body weight, growth rate, and survival rate of shrimps, as well as disease resistance against *V. parahaemolyticus* infection (34). Another study showed that adding *C. butyricum* CB to the basal diet improved the intestinal structure (villus height and epithelial cell length) and regulated the balance of intestinal microbiota, thus improving the growth of *Larimichthys crocea* larvae (35). In addition, Luo et al. (32) showed that *C. butyricum*-supplemented diets stimulated the activity of antioxidant enzymes and intestinal tissue health, as well as increasing the survival rate of juvenile grass turtle (*C. reevesii*) after infection with *Aeromonas veronii*. These results suggested the potential of *C. butyricum* in aquatic animals, but most bacterial strains originated from other sources rather than the gut of aquatic animals (22). In addition, the isolation of *C. butyricum* from mud crabs had not been achieved yet. In this study, the butyrate-producing bacterium *C. butyricum* G13 was isolated for the first time from the intestinal tract of mud crabs. It was further characterized by evaluating its tolerance to gastrointestinal conditions (pH, NaCl, and bile salts), hydrophobicity, ability to degrade starch and cellulose, sensitivity to antibiotics, and ability to inhibit the growth of pathogens. In addition, the influence of *C. butyricum* G13 alone or in combination with prebiotics (galactooligosaccharides [GOS] and/or resistant starch [RS]) on the production of SCFAs and the intestinal microbiota was investigated using *in vitro* fermentation. Additionally, the protective role of *C. butyricum* G13 alone or in combination with both GOS and RS (G13+GOS+RS) in mud crabs against bacterial infections was studied. The results of this study demonstrated that *C. butyricum* G13 can be considered a potential probiotic for use in mud crab aquaculture.

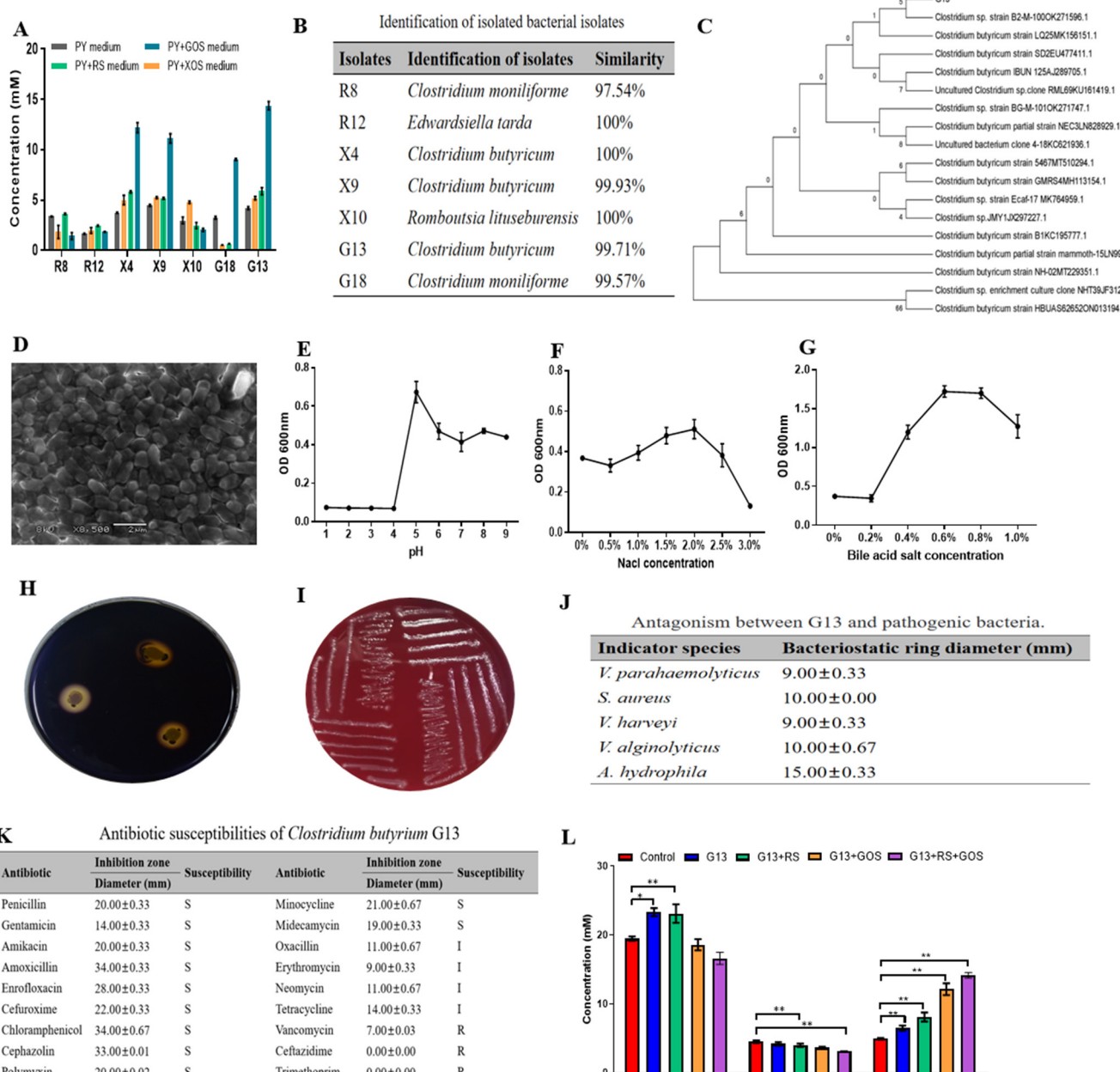

**FIG 1** Isolation, identification, and *in vitro* characterization of *C. butyricum* G13. (A) Production of butyric acid in PY broth incubated with different bacterial strains (including *C. butyricum* G13). (B) Identification of butyrate-producing bacteria based on the 16S rRNA sequences and comparison with available database sequences using BLASTN search. (C) Neighbor-joining phylogenetic tree showing the relationship between *C. butyricum* G13 and other species (accession numbers in the parentheses), generated by MEGA 5.2.2 software; the numbers at the branches represent bootstrap support values (1,000 replications). (D) Morphology of *C. butyricum* G13 observed under scanning electron microscopy. (E to G) Growth of *C. butyricum* G13 at different pH, NaCl, and bile salt levels. (H and I) and Amylase activity and hemolytic test, respectively. (J) Antimicrobial activity of *C. butyricum* G13 against pathogenic bacteria. (K) Antibiotic susceptibility of *Clostridium butyricum* G13. (L) Short-chain fatty acid (SCFA) production in PY broth supplemented with either *C. butyricum* G13 (G13), G13 and resistant starch (G13+RS), G13 and GOS (G13+GOS), or G13, resistant starch, and GOS (G13+RS+GOS) after *in vitro* fermentation of mud crab gut contents. Comparisons of differences in each group combined with *t* test analysis ($P < 0.05$) and the significant differences are indicated with an asterisk (*, $P < 0.05$; **, $P < 0.01$).

## RESULTS

**Isolation and identification of bacteria.** A total of 7/102 bacterial strains isolated from the gut contents of mud crabs produced a high concentration of butyrate (>2 mM) (Fig. 1A). The 16S rRNA sequence was compared with available database sequences using BLASTN search, and the results are shown in Fig. 1B. The BLASTN search and phylogenetic analysis revealed that G13 was highly similar to the species

*Clostridium butyricum* (accession number NR_113244.1) (99.85%) and identified as *C. butyricum* G13 (GenBank accession number OQ607756). Phylogenetic tree analysis showed that *C. butyricum* G13 formed a separate cluster with *Clostridium* spp. (Fig. 1C). Moreover, the results showed that among the obtained bacterial isolates, *C. butyricum* G13 produced the highest butyrate concentration, so it was selected as a presentative strain for further investigations. *C. butyricum* G13 is a Gram-positive bacterium, with short rod shape (1 to 2 $\mu$m), and the colony was round, white, and moist and grew rapidly on peptone-yeast extract (PY) agar (30 mm in diameter after 2 days of culture). The morphological characteristics of *C. butyricum* G13 are shown in Fig. 1D.

Biological characteristics of *C. butyricum* G13 were determined, and the results showed that the bacterium can grow in a wide range of pHs 4 to 9, NaCl concentrations (1 to 2.5%), and bile salt concentrations (0.2 to 1.0%) (Fig. 1E to G). The bacterium exhibited a strong amylase activity, with a zone of inhibition at 14 mm in diameter (Fig. 1H). The results of hemolytic activity showed that *C. butyricum* G13 is nontoxic, and the activity was classified as gamma-hemolysis (Fig. 1I). *C. butyricum* G13 revealed a high hydrophobicity, 43.8% (the optical densities at 560 nm [$OD_{560}$] of the bacterial suspension with and without xylene were 1.345 and 0.907, respectively), indicating the ability of strong adhesion of the bacterium to the intestinal tract. The results of the antimicrobial assay revealed that *C. butyricum* G13 inhibited the growth of the pathogens *V. parahaemolyticus*, *Staphylococcus aureus*, *Vibrio harveyi*, *V. alginolyticus*, and *Aeromonas hydrophila*, with an inhibition zone diameter of >9 mm (Fig. 1J). The antibiotic susceptibility results showed that *C. butyricum* G13 was sensitive to penicillin, ampicillin, amikacin, amoxicillin, enrofloxacin, cefuroxime, chloramphenicol, cephazolin, gentamicin, midecamycin, polymyxin, and minocycline; it was intermediate to oxacillin, erythromycin, neomycin, and tetracycline and resistant to ceftazidime, vancomycin, and trimethoprim (Fig. 1K).

**Short-chain fatty acids produced by *in vitro* fermentation.** In the *in vitro* anaerobic fermentation experiment, the effects of *C. butyricum* G13 either alone or in combination with prebiotics (GOS and/or RS) on the metabolic activity of the intestinal microbiota in mud crabs were studied. After 24 h of fermentation, the content of SCFAs (including acetic, propionic, and butyric acids) was significantly different from that of the control (Fig. 1L). The results showed that acetic acid accounted for the highest content of total SCFAs, followed by butyric acid and propionic acid. The concentration of acetic acid in the G13 group was the highest, which was significantly higher than in the control ($t = 5.813$ and $P = 0.00$), G13+GOS ($t = 4.745$ and $P = 0.00$), and G13+GOS+RS ($t = 6.325$ and $P = 0.00$) groups. The propionic acid content in the G13+GOS+RS group ($t = 7.452$ and $P = 0.00$) was significantly lower than in the control, whereas the G13 ($t = 0.9538$ and $P = 0.36$), G13+RS ($t = 1.795$ and $P = 0.1$), and G13+GOS ($t = 3.633$ and $P = 0.00$) groups were similar to the control. The level of butyric acid in the G13 ($t = 3.831$ and $P = 0.00$), G13+RS ($t = 4.731$ and $P = 0.00$), G13+GOS ($t = 8.389$ and $P = 0.00$), and G13+GOS+RS ($t = 21.98$ and $P = 0.00$) groups were significantly higher than that in the control.

**Gut microbiota sequencing and microbial diversity after *in vitro* fermentation.** The effects of *C. butyricum* G13 alone or in combination with prebiotics (GOS and/or RS) on changes in gut microbial community structure were examined using the *in vitro* fermentation culture. The saturation of the sequencing results was reflected by the saturation platform of the sparse curve (Fig. 2A). The similarity of the diversity of all samples was shown by the rank abundance curve (Fig. 2B). High-throughput sequencing analysis of the V3-V4 region of the 16S rRNA gene showed that a total of 6,945,322 high-quality reads were obtained from 33 samples. The raw data of 16S rRNA sequences was deposited in the GenBank under accession number PRJNA948684. The number of reads ranged from 84,726 to 91,712, with a distribution of those in the control of 77,434 to 93,886, the G13 group of 72,233 to 96,670, the G13+GOS group of 77,741 to 92,811, the G13+RS group of 84,7771 to 96,314, and the G13+GOS+RS group of 82,403 to 90,216. The Good coverage was >99% in all samples, indicating that the bacterial community in each sample was basically identified, which could reach the saturation depth of sequencing. The number of shared operational taxonomic units (OTUs) in all treatment groups

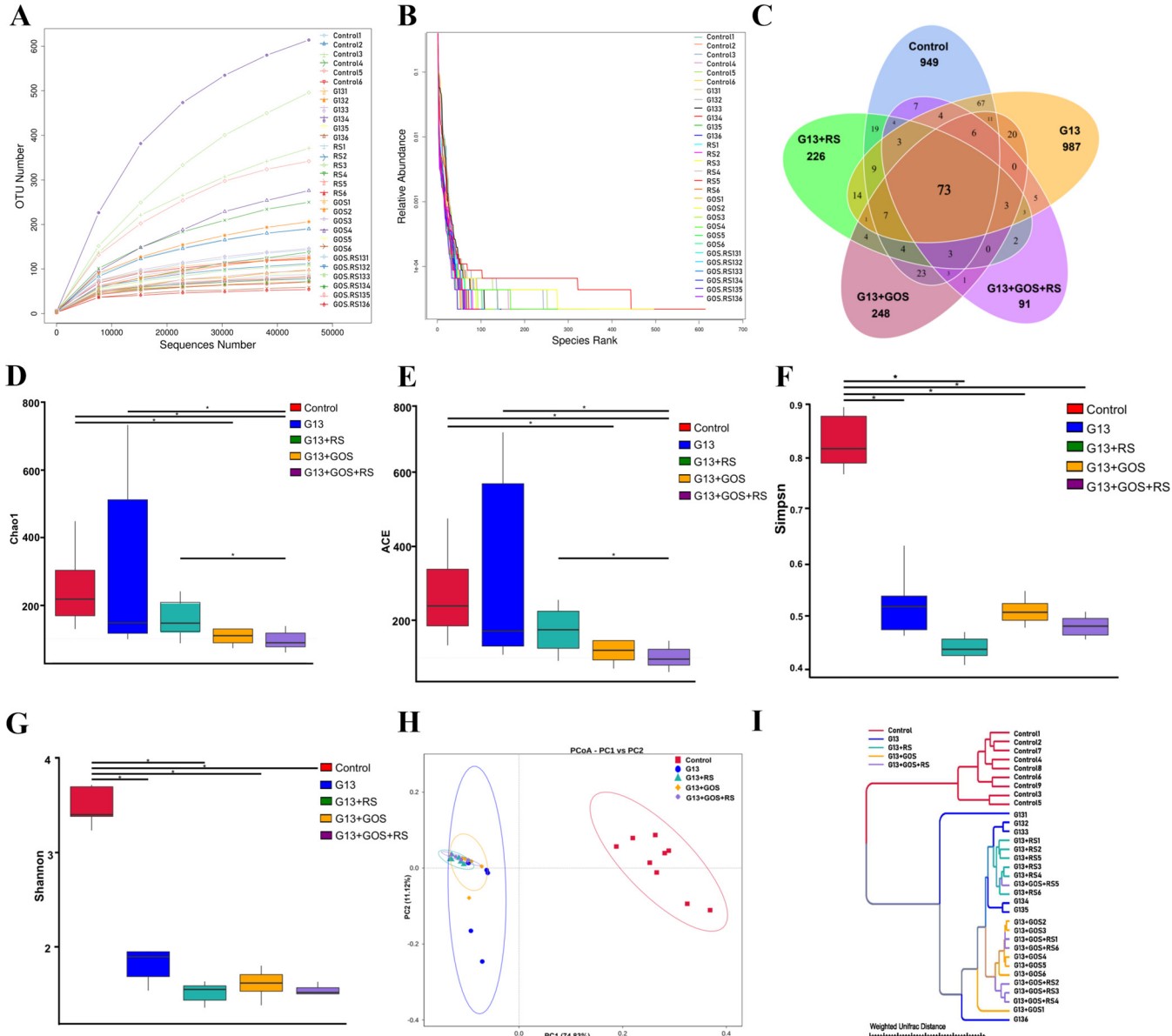

**FIG 2** Diversity and composition of gut microbiota in the *in vitro* anaerobic fermentation supplemented with either G13, G13+RS, G13+GOS, or G13+RS+GOS. (A and B) Rarefaction curves represent the richness of the gut microbiota in samples of each group. (C) Number of shared and unique OTUs between groups. (D to H) Diversity and richness of the gut microbiota in groups (based on the indexes of Chao1, ACE, Simpson, and Shannon). (I) PCoA based on the weighted UniFrac distance was used to represent the community structure similarity between groups, and the unweighted pair group method using average linkages (UPGMA) cluster tree structure represented the cluster distribution between samples. Comparison of differences in each group combined with *t* test analysis ($P < 0.05$) and the significant differences ($P < 0.05$) are indicated with an asterisk.

was 73, and the numbers of unique OTUs in the control, G13, G13+RS, G13+GOS, and G13+RS+GOS groups were 949, 987, 226, 248, and 91, respectively (Fig. 2C).

The alpha diversity of intestinal bacterial communities in the cultures after *in vitro* fermentation was estimated by the Chao1, ACE, Shannon, and Simpson indices. The results showed that the Chao1 and ACE indices in the G13 ($t = 0.087$ and $P = 0.93$ for Chao1 and $t = 0.002$ and $P = 0.998$ for ACE), G13+GOS ($t = 2.882$ and $P = 0.01$ for Chao1 and $t = 3.065$ and $P = 0.01$ for ACE), G13+RS ($t = 3.052$ and $P = 0.01$ for Chao1 and $t = 3.298$ and $P = 0.008$ for ACE), and G13+GOS+RS ($t = 5.021$ and $P = 0.00$ for Chao1 and $t = 5.645$ and $P = 0.00$ for ACE) were significantly higher than that in the control. Shannon and Simpson indices were significantly decreased in the G13 ($t = 7.862$ and $P = 0.00$ for Shannon and $t = 9.0570$ and $P = 0.00$ for Simpson), G13+GOS ($t = 18.06$ and $P = 0.00$ for Shannon and $t = 13.69$ and $P = 0.00$ for Simpson), G13+RS ($t = 20.84$

and $P = 0.00$ for Shannon and $t = 17.09$ and $P = 0.00$ for Simpson), and G13+GOS+RS ($t = 20.67$ and $P = 0.00$ for Shannon and $t = 15.50$ and $P = 0.00$ for Simpson) groups compared to the control (Fig. 2D to H).

Beta diversity analysis based on permutational multivariate analysis of variance (PERMANOVA) and analysis of similarity (ANOSIM) showed that the overall bacterial community structure of the control was significantly different from that of other groups (Bonferroni-corrected $P < 0.05$). Principal-coordinate analysis (PCoA) and weighted pair group algorithm based on weighted UniFrac distance matrix (WPGMA) were used to estimate the similarity of microbial community composition. The results showed that the gut microbiota of the G13, G13+RS, G13+GOS, and G13+GOS+RS groups basically clustered together and were far away from the control (Fig. 2I).

**Gut microbiota composition after *in vitro* fermentation.** Differences in gut microbiota in groups after 24 h of *in vitro* fermentation were investigated at both the phylum and genus levels (Fig. 3). The top 10 phyla in all groups are shown in Fig. 3A. The results showed that *Proteobacteria* and *Firmicutes* were the most predominant. The relative abundance of *Proteobacteria* was higher in the G13 ($t = 17.68$ and $P = 0.00$), G13+RS ($t = 23.00$ and $P = 0.00$), G13+GOS ($t = 20.60$ and $P = 0.00$), and G13+GOS+RS ($t = 19.43$ and $P = 0.00$) groups than in the control, whereas that of *Firmicutes* was significantly lower in the G13 ($t = 19.90$ and $P = 0.00$), G13+RS ($t = 23.18$ and $P = 0.00$), G13+GOS ($t = 20.66$ and $P = 0.00$), and G13+GOS+RS ($t = 19.39$ and $P = 0.00$) groups.

The relative abundances of the top 10 bacterial genera in different groups are shown in Fig. 3B. The relative abundances of *Edwardsiella* and *Clostridium sensu stricto* 1 were significantly higher in the G13 ($t = 26.19$ and $P = 0.00$ for *Edwardsiella* and $t = 1.383$ and $P = 0.196$ for *Clostridium sensu stricto* 1), G13+RS ($t = 50.38$ and $P = 0.00$ for *Edwardsiella* and $t = 6.164$ and $P = 0.00$ for *Clostridium sensu stricto* 1), G13+GOS ($t = 10.00$ and $P = 0.00$ for *Edwardsiella* and $t = 10.00$ and $P = 0.00$ for *Clostridium sensu stricto* 1), and G13+GOS+RS ($t = 43.75$ and $P = 0.00$ for *Edwardsiella* and $t = 12.12$ and $P = 0.00$ for *Clostridium sensu stricto* 1) groups than in the control. In contrast, *Proteocatella* was found to be significantly lower in the G13 ($t = 4.412$ and $P = 0.001$), G13+RS ($t = 6.382$ and $P = 0.00$), G13+GOS ($t = 7.032$ and $P = 0.00$), and G13+GOS+RS ($t = 7.223$ and $P = 0.00$) groups than in the control.

Linear discriminant analysis effect size (LEfSe) analysis (with linear discriminant analysis [LDA] of >3.0) was also used to investigate the involvement of the specific bacterial taxa in groups (Fig. 3C and D). The results showed that the control group was enriched with *Peptostreptococcales*, *Peptostreptococcaceae*, *Firmicutes*, *Clostridia*, and *Proteocatella*. In the G13 group, microbial terms such as mitochondria, *Rickettsiales*, chloroplasts, cyanobacteria, and *Clostridium* were significantly enriched. In the G13+GOS group, *Clostridium butyricum*, *Clostridiaceae*, *Lactococcus*, and *Streptococcaceae* were predominant. The taxa *Hafniaceae*, *Enterobacterales*, and *Gammaproteobacteria* were significantly enriched in the G13+RS group and the taxon *Lactococcus* was enriched in the G13+GOS+RS group.

**Functional profile of intestinal microbiota after *in vitro* fermentation.** The function profile of the intestinal microbiota with respect to the supplementation of *C. butyricum* G13 either alone or in combination with prebiotics (GOS or/and RS) was analyzed after *in vitro* anaerobic fermentation using PICRUSt analysis (Fig. 4). The principal-component analysis (PCA) results showed that the functional composition in the G13, G13+RS, G13+GOS, and G13+GOS+RS groups was significantly different from that in the control (Fig. 4A). The distribution of unique and shared genes in different experimental groups is shown in the Venn diagram in Fig. 4B. There were 4,479 shared genes in all groups, as well as 56, 11, 47, and 249 unique genes in the control, G13, G13+GOS, and G13+GOS+RS groups, respectively. The functional differences among the groups were analyzed and are exhibited in Fig. 4C. Figure 4C shows that the "folding, sorting and degradation," "metabolism of other amino acids," "lipid metabolism," "endocrine system," "glycan biosynthesis and metabolism," "cellular functions and processes," and "signaling transduction and membrane transport" were significantly enriched in the G13, G13+GOS, and G13+GOS+RS compared to the control. However, the functions of "bacterial motility proteins," "ribosome biogenesis," and "cysteine and

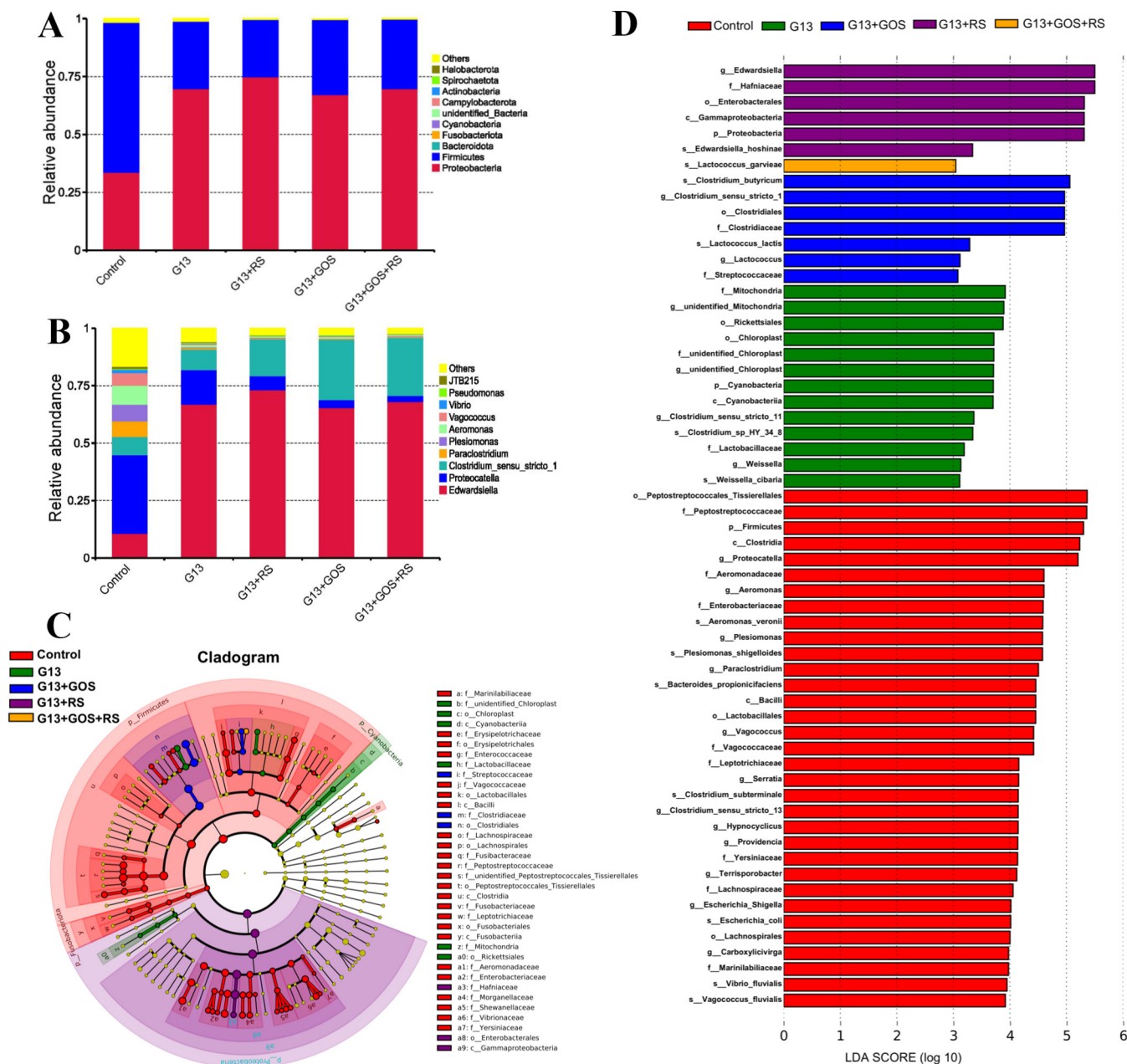

**FIG 3** Composition of the gut microbial community in experimental groups (control, G13, G13+RS, G13+GOS, and G13+RS+GOS) after *in vitro* anaerobic fermentation. (A and B) Relative abundance of gut microbiota in groups at the phylum (A) and genus (B) levels. (C) LEfSe analysis of the phylogenetic distribution of microbial taxa among groups. (D) Histogram of LDA value distribution representing the significant abundance of microbiota among groups (with LDA scores of >3.0).

methionine metabolism, and others" in the G13, G13+GOS, and G13+GOS+RS groups were significantly less abundant than in the control.

**C. butyricum G13 protected the mud crab against bacterial infection *in vivo*.** The role of *C. butyricum* G13 either alone or in combination with both GOS and RS in protecting mud crab resistance to *V. parahaemolyticus* infection was investigated (Fig. 5). After 10 days of challenge, the survival rate of mud crabs was significantly improved in the G13+*V. parahaemolyticus* (63.33%) ($t = 5.612$ and $P = 0.00$) and G13+RS+GOS+*V. parahaemolyticus* (86.67%) ($t = 4.353$ and $P = 0.01$) groups compared with the *V. parahaemolyticus* (53.33%) group (Fig. 5A). The effect of G13 alone or in combination with both GOS and RS on the production of SCFAs after infection with *V. parahaemolyticus* was also studied (Fig. 5B). As shown in Fig. 5B, the total amount of SCFAs, especially acetic and butyric

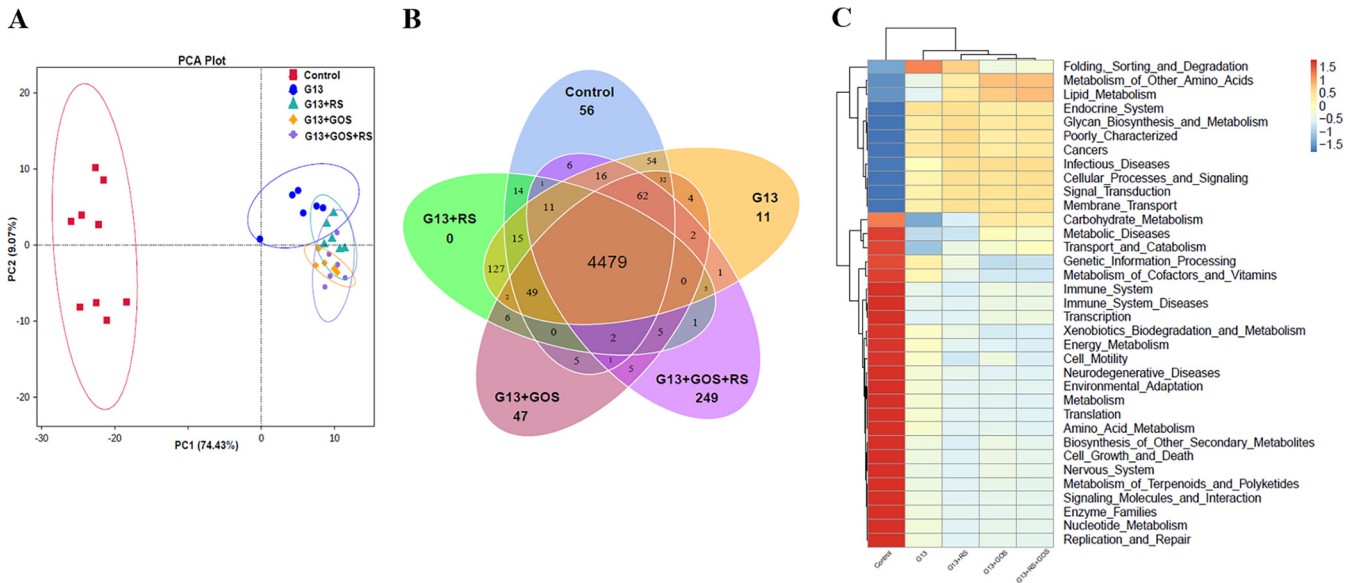

**FIG 4** Prediction of the gut microbial function in experimental groups (control, G13, G13+RS, G13+GOS, and G13+RS+GOS) after *in vitro* anaerobic fermentation, rendered by PICRUSt. (A) The PCA reduced-dimension diagram shows the functions of the gut microbiota in each group. (B) Distribution of gene number among groups. (C) Relative abundance and signal pathways of gut microbiota in experimental groups were analyzed functionally (at level 2).

acids, in the G13+*V. parahaemolyticus* and G13+GOS+RS+*V. parahaemolyticus* groups was significantly higher than in other groups. Compared with that in the control, the acetic acid content in the *V. parahaemolyticus* group was significantly decreased ($t$ = 2.533 and $P$ = 0.03), the propionic acid content in the G13+*V. parahaemolyticus* and G13+GOS+RS+*V. parahaemolyticus* groups was significantly increased ($t$ = 3.802 and $P$ = 0.00 for the former and $t$ = 2.282 and $P$ = 0.00 for the latter), and the butyric acid content in the G13+GOS+RS+Vp was significantly increased ($t$ = 4.615 and $P$ = 0.00). Moreover, the content of acetic and butyric acids was significantly increased in the G13+GOS+RS+*V. parahaemolyticus* group compared to that in the *V. parahaemolyticus* group ($t$ = 4.537 and $P$ = 0.00 for the former and $t$ = 4.695 and $P$ = 0.00 for the latter). Furthermore, the histological analysis (hematoxylin and eosin [H&E] staining) revealed that the normal gut structure was observed in the G13+*V. parahaemolyticus*, G13+GOS+RS+*V. parahaemolyticus*, and control groups, whereas severe gut structural damage was present in the *V. parahaemolyticus* group (Fig. 5C to F). Additionally, the activity of the enzymes superoxide dismutase (SOD), catalase (CAT), alkaline phosphatase (AKP), and acid phosphatase (ACP) enzymes was significantly increased, but malonaldehyde (MDA) activity was significantly decreased in the hepatopancreas of mud crabs (in the G13 and G13+GOS+RS groups) after challenge with *V. parahaemolyticus* (Fig. 5G to K). These results indicate that G13 and G13+GOS+RS can improve the immune response of mud crabs, subsequently enhancing resistance to *V. parahaemolyticus* infection.

## DISCUSSION

Previous studies have shown that the changes in the intestinal microbiota and probiotics play an important role in the healthy development of the host. Recently, probiotics that produce butyrate have been widely used in aquaculture (36). Microbial community composition and metabolism are significantly related to the content and composition of SCFAs in the intestine, especially butyric acid (37–39). *C. butyricum* produces butyric acid, which can improve the balance of the host intestinal flora and reduce the probability of intestinal infection with pathogenic bacteria (40). In this study, the bacterial strain *C. butyricum* G13 produced a high butyric acid content isolated from the intestinal contents of the mud crab, and its probiotic potential was evaluated. The results showed that *C. butyricum* G13 tolerates a wide arrange of conditions of the gastrointestinal tract, including pHs of 4 to 9, NaCl concentrations of 1.0 to 2.5%, and bile salt concentrations

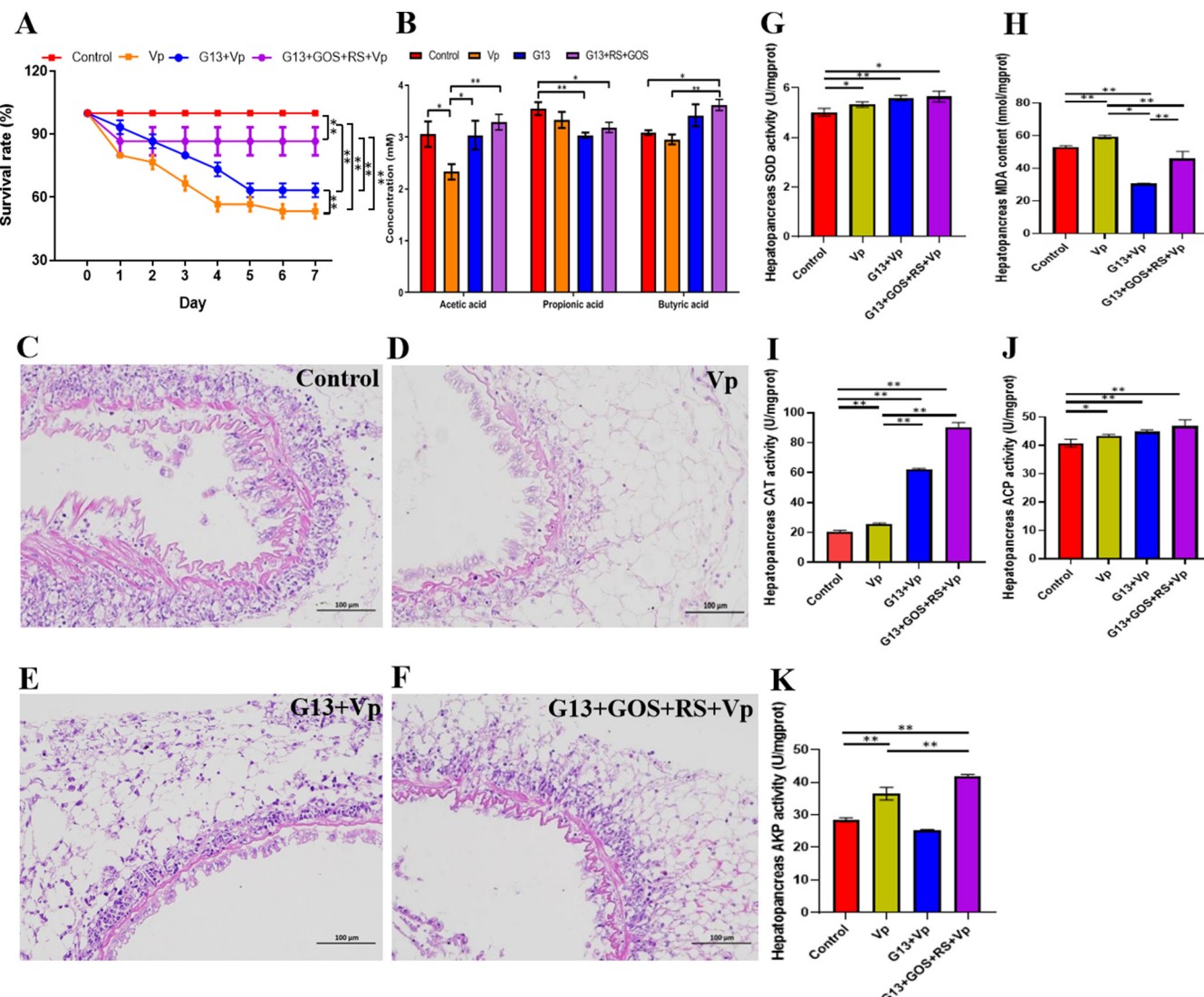

**FIG 5** Results of short-chain fatty acid (SCFA) production and physiological responses of mud crabs after infection with *Vibrio parahaemolyticus*. (A) Survival rate of mud crabs in experimental groups (control, *V. parahaemolyticus* [Vp], G13+Vp, and G13+RS+GOS+Vp). (B) SCFAs produced in the gut contents of mud crabs in different experimental groups. (C to F) Structure of midgut of mud crabs in different experimental groups (H&E stain). (G to K) Activity of enzymes (SOD, MDA, CAT, AKP, and ACP) in the hepatopancreas of mud crabs in different experimental groups. Comparison of differences in each group combined with *t* test analysis and the significant differences are indicated with an asterisk (*, $P < 0.05$; **, $P < 0.01$).

of 0.2 to 1%. This indicated that the bacterium can colonize and grow in the intestinal environment. The findings reported here are similar to those for *Bacillus subtilis* isolated from grass carp (*Ctenopharyngodon idellus*); the results showed that bacterial spores have a high tolerance to low pH (1 to 3) and high bile concentrations (2 to 10%) (41). In another study, the results showed that *Lactobacillus plantarum* is able to live in a low pH (2.5) and 0.3% bile salt (42). In this study, we found that *C. butyricum* G13 shows high sensitivity to 15/19 antibiotics, indicating that the bacterium has a low risk of horizontal transfer of antibiotic resistance genes between bacteria (40, 43). Moreover, the results showed that *C. butyricum* G13 can inhibit the growth of pathogenic bacteria (*V. parahaemolyticus*, *S. aureus*, *V. harveyi*, *V. alginolyticus*, and *A. hydrophila*). This is consistent with a report by Gao et al. (33), which showed inhibition of the growth of *Streptococcus enteritidis* and *V. parahaemolyticus* and their adhesion to carp intestinal epithelial cells by the inclusion of *C. butyricum* MIYAIRI II588. Another study showed that *C. butyricum* CB2 also has significant antagonistic effects on the growth of pathogenic bacteria, comprising *A. hydrophila* and *Vibrio anguillarum* (28). Taken together, the outcomes of this study indicate the probiotic potential of *C. butyricum* G13.

Butyric acid is the end product of microbial metabolism, and its synthesis is mainly dependent on the presence of butyrate-producing bacteria and carbohydrates in the intestinal tract (22). It has been reported that butyric acid can be a main energy source for animal intestinal epithelial cells, promote the proliferation, differentiation, growth, and migration of intestinal cells, regulate the diversity and composition of gut microbiota, have anti-inflammatory effects, and improve host resistance against infection (44, 45). In this investigation, in the *in vitro* fermentation, the results showed that *C. butyricum* G13 alone or in combination with GOS and/or RS significantly increased the production of SCFAs, especially butyric acid. This is in agreement with previous findings that dietary supplementation of *C. butyricum* can increase the SCFAs in the intestine of the Pacific white shrimp (*Litopenaeus vannamei*) (29) and kuruma shrimp (*Marsupenaeus japonicus*) (46). This indicated that the addition of *C. butyricum* G13 in this study may have increased the level of total SCFAs in the intestinal tract of mud crabs and exhibited protective effects on their host. However, the issue needs to be more investigated in an *in vivo* study.

*C. butyricum* supplementation can modify the balance of the intestinal microbiota in aquatic animals (47, 48). The results of this study revealed that *C. butyricum* G13 had significant impacts on the structure of the intestinal microbiota of mud crabs based on the alpha diversity parameters and PCoA. It is noted that the values of Chao1 and ACE indices of *C. butyricum* G13 were increased compared to those of the control. This is similar to the previous report that largemouth bass fed with *C. butyricum*-fermented Chinese herbal medicine showed an abundance of both *Proteobacteria* and *Firmicutes* (49). In this study, we found that the composition, structure, and function of the intestinal microbiota of mud crabs were also affected by the supplementation of *C. butyricum* G13 alone or in combination with GOS and/or RS. Compared with that of the control group, the relative abundance of *Proteobacteria* was significantly increased while that of *Firmicutes* was decreased in the G13, G13+RS, G13+GOS, and G13+GOS+RS groups. The experimental results differ from the previous reports of Li et al. (50), which showed that the abundance of *Bacteroidetes* and *Firmicutes* was increased by feeding the tilapia with the dietary addition of *C. butyricum* CB2. Our study also found that the increase in the relative abundance of *Proteobacteria* was related to an increase of the genus *Edwardsiella*. Previous studies have demonstrated that the members of the genus *Edwardsiella* are generally considered pathogens of aquatic animals (51, 52). However, in a recent study, the heat-inactivated *Edwardsiella* sp. strain 34HN has been shown to be a probiotic bacterium, which could improve growth performance and enhance the immune response of African catfish (*Clarias gariepinus*) during 100 days of feeding (53). In addition, an increase in the relative abundance of *Clostridium sensu stricto* 1 was found with *C. butyricum* G13 alone or in combination with GOS and/or RS, indicating that supplementation can promote the growth of the members of butyrate-producing bacteria, including *Clostridium*, and show the direct relationship between *C. butyricum* G13 and butyric acid production. This is consistent with the case of *C. butyricum* CB1 administered to black tiger shrimp (*P. monodon*) (31). Therefore, the outcomes of this study indicate the effects of *C. butyricum* G13 on the homeostasis of the intestinal microbiota of mud crabs *in vitro*.

Interestingly, the previous *in vitro* study using healthy crucian carp FIECs found that *C. butyricum* and its supernatant can significantly inhibit the growth and colonization of the pathogens *S. enteritidis* and *V. parahaemolyticus* (33). This is worthwhile for further *in vivo* investigations. Interestingly, the results of our *in vivo* study found that a higher survival rate and maintenance of the normal structure of the gut morphology were recorded during *V. parahaemolyticus* infection. Thus, the administration of *C. butyricum* G13 alone or in combination with both GOS and RS can protect mud crabs against infection. This is consistent with the previous study by Luo et al. (32), which showed that the supplementation of different forms (live cells, sonication-killed cell extracts, and fermentation supernatant) of *C. butyricum* improved the cumulative survival of Pacific white shrimps after challenge with *V. parahaemolyticus*. In addition, the supplementation of *C. butyricum* G13 alone or in combination with both GOS and RS

(G13+GOS+RS) significantly increased the activity of SOD, CAT, AKP, and ACP enzymes and decreased that of MDA enzyme in the hepatopancreas of mud crabs. These results indicate that G13 alone or combined with prebiotics can protect mud crabs against *V. parahaemolyticus* infection through stimulation of the immune responses. In a previous study, Sun et al. (54) fed giant freshwater prawns (*Macrobrachium rosenbergii*) with *C. butyricum* for 8 weeks and found that *C. butyricum* significantly increased the SOD but decreased the MDA levels. The expression of peroxiredoxin 5 and Toll was upregulated under ammonia stress to enhance the growth performance of the host. This is consistent with the results of this study. Thus, *C. butyricum* G13 has probiotic potential in improving mud crab health.

In this study, *C. butyricum* G13 was isolated for the first time from the gut of mud crabs and identified as a potential probiotic bacterium. *In vitro* fermentation revealed that the addition of *C. butyricum* G13 alone or in combination with GOS and/or RS can promote the production of butyric acid and modulate the gut microbiota. Furthermore, the results of *in vivo* study revealed the importance of both *C. butyricum* G13 and its combination with GOS and RS in protecting mud crabs against *V. parahaemolyticus* infection. Taken together, our findings show that *C. butyricum* G13 can be used as a probiotic bacterium in mud crab aquaculture. However, further studies regarding the effects of dietary supplementation of *C. butyricum* G13 on growth, physiological changes, and the gut microbiota of mud crabs as well as the molecular mechanisms by which *C. butyricum* G13 protects mud crabs against *V. parahaemolyticus* infection need to be considered.

## MATERIALS AND METHODS

**Isolation and identification of bacteria.** Eighteen healthy mud crabs (body weight, appropriately 43.3 g) were purchased from a culture farm in Niutianyang (Shantou, Guangdong, China) and randomly divided into three groups. Sterile dissection of mud crabs was carried out on ice, the intestine was aseptically collected, and the gut contents from the posterior intestine were gently scraped, placed into sterile Eppendorf tubes, and quickly transferred to an anaerobic workstation (10% $H_2$, 5% $CO_2$, 85% $N_2$; Shanghai Longyue, China). The gut contents were thoroughly mixed in a 10-fold dilution of an anaerobic 0.1 M sodium phosphate buffer (pH 6.8). A volume of 0.1 mL was incubated in Hungate tubes sealed with butyl rubber stoppers and screw caps of $O_2$-free $CO_2$-PY broth (5.0 g of peptone, 5.0 g of trypsin, 1 mg of vitamin $K_1$, 10.0 g of yeast extract, 0.5 g of cysteine, 4.0 g of $NaCO_3$, 10 mL of 0.05% heme solution, 0.4 g of $K_2HPO_4$, 0.04 g of $KH_2PO_4$, 0.08 g of $NaHCO_3$, 0.04 g of NaCl, 8 mg of $CaCl_2$, 1.9 $\mu$g of $MgSO_4 \cdot 7H_2O$ [pH 6.8]) (5 mL) supplemented with 0.05 g of GOS or RS (55). The culture tubes were incubated on a shaker (Shanghai Bluepard Instruments Co., Ltd.) at $140 \times g$ for 24 h at 28°C and then used for the isolation of bacteria. After a 24-h incubation, the stock solution was diluted to $10^{-6}$, $10^{-7}$, and $10^{-8}$ serial dilutions, and a 100-$\mu$L suspension was spread onto PY agar plates and cultured in an anaerobic chamber at 28°C for 24 h. The colonies with different colors and shapes were selected and transferred to new PY agar plates. The colonies were purified 4 to 5 times until they grew into single colonies. The single colonies were transferred to PY broth alone or with supplementation with GOS or RS and shaken at $200 \times g$ for 24 h at 28°C. The cultures were used for the determination of butyric acid production by gas chromatography, and the butyrate-producing bacterial isolates were identified with a net value of >2 mM butyric acid production (56). The morphology of the bacterial isolates was observed by scanning electron microscopy (JSM-6360LA, Jeol, Japan).

The genomic DNA of isolates was extracted using the TIANamp genomic DNA kit (Tiangen Biotech Co., Ltd., Beijing, China) following the manufacturer's instructions. The extracted DNA was used for the PCR amplification of the 16S rRNA gene using universal primers 27F/1492R. The amplification program was 95°C for 5 min, 94°C for 30 s, and 55°C for 1 min, followed by 30 cycles of 72°C for 2 min and then 72°C for 5 min. The amplified samples were detected by 1% agarose gel electrophoresis. The PCR products were used for 16S rRNA sequencing by a commercial company. The identity of gene sequences was determined using BLASTN search in the GenBank database.

**Short-chain fatty acid analysis.** The bacterial culture was inoculated into PY broth, shaken, and incubated at $200 \times g$ for 24 h at 28°C. At 0 h and 24 h, 2 mL of culture solution was sampled by sterile syringe. The samples were centrifuged at $12,000 \times g$ for 10 min at 4°C, and the supernatants were retained. The supernatants were adjusted to pH 2 to 3 by adding $H_2SO_4$, centrifuged at $12,000 \times g$ for 10 min at 4°C, and collected using a 0.22-$\mu$m filter membrane. SCFA determination was performed using the Agilent GC6890N network gas chromatograph (Agilent Technologies, Santa Clara, CA, USA) with HP-InnoWax capillary columns. Nitrogen (purity, >99.99%) was used as carrier gas and tail-blowing gas with flow rates of 1.0 mL/min and 45 mL/min, respectively. The hydrogen gas flow rate was 40 mL/min, the airflow rate was 450 mL/min. The instrument was initially operated at 50°C for 6 min and then at 230°C for 8 min. Each injection volume was 0.5 $\mu$L; the flame ionization detection (FID) temperature was 250°C.

The standard curve of SCFAs was prepared by an external standard method. The mixed standard sample solution of acetic, propionic, and butyric acids was made. The concentration of acetic acid was adjusted to 0.089 to 22.876 mM, propionic acid was 0.117 to 29.931 mM, and butyric acid was 0.143 to

36.560 mM. Each sample was set up in triplicate. The calibration curves were prepared by plotting the relative peak area to the molarity of the solution. Concentrations of acetic, propionic, and butyric acids were determined using the standard curve of each SCFA and expressed as mean millimolar concentration.

***In vitro* characterization of isolated bacteria.** To determine tolerance to pH, NaCl, and bile salt resistance, the bacteria were cultured in PY broth for 24 h at 28°C by shaking at $200 \times g$. After a 24-h incubation, the bacterial density was adjusted to $10^7$ CFU/mL, and 100 $\mu$L of the bacterial suspension was transferred to PY broth adjusted to different values of pH (1, 2, 3, 4, 5, 6, 7, and 8), bile salt (0, 0.2, 0.4, 0.6, 0.8, and 1%), or NaCl (0, 0.5, 1, 1.5, 2, 2.5, and 3%). Each treatment was performed in three replications. The incubation was shaken at $200 \times g$ for 24 h at 28°C. The absorbance was detected by optical density (OD) at 600 nm to determine the growth of the bacteria.

**Antimicrobial activity and antibiotic susceptibility testing.** The antimicrobial activity was performed using the disk agar diffusion method. The bacterial isolate (G13) was grown overnight in PY broth for 24 h at 28°C with shaking at $200 \times g$. The cultures were centrifuged at $12,000 \times g$ for 10 min at 4°C, and the supernatants were obtained and used for the antimicrobial activity tests. The pathogenic bacteria *V. parahaemolyticus*, *Staphylococcus aureus*, *Vibrio harveyi*, *Vibrio alginolyticus*, and *Aeromonas hydrophila* were used in this assay. The bacteria *V. parahaemolyticus*, *S. aureus*, and *V. harveyi* were cultured in LB medium (37°C), while *V. alginolyticus* and *A. hydrophila* were in tryptic soy broth (TSB) medium (28°C) for 24 h with shaking at $200 \times g$. A 100-$\mu$L bacterial suspension was first spread onto the agar plates, and the paper disc (6 mm in diameter) wetted by the supernatants from the *C. butyricum* G13 culture was laid on the agar plate surface. Each treatment was performed in three replications. The plates were cultured for 24 h at 28°C and the appearance of the inhibition zone was observed.

The antibiotic susceptibility test was performed using the agar disc diffusion method. A total of 19 antimicrobial agents, including penicillin (10 $\mu$g), oxacillin (1 $\mu$g), ampicillin (10 $\mu$g), amikacin (30 $\mu$g), amoxicillin acid (30 $\mu$g), enrofloxacin (10 $\mu$g), cefuroxime (30 $\mu$g), chloramphenicol (10 $\mu$g), cephazolin (30 $\mu$g), ceftazidime (30 $\mu$g), erythromycin (15 $\mu$g), gentamicin (10 $\mu$g), midecamycin (30 $\mu$g), polymyxin B (300 $\mu$g), neomycin (30 $\mu$g), minocycline (10 $\mu$g), tetracycline (30 $\mu$g), vancomycin (30 $\mu$g), and trimethoprim (30 $\mu$g), were used for this study. A volume of 100 $\mu$L of bacterial solution ($1 \times 10^8$ CFU/mL) was spread onto PY agar plates and the antimicrobial agent discs (Hangzhou Microbial Reagent Factory, Hangzhou, China) were laid on the agar surface. Each treatment was carried out three times. The incubation was conducted in an anaerobic workstation for 24 h at 28°C, and the diameters of bacteriostatic circles were measured.

**Digestive enzyme detection.** The enzymatic degradation of starch of the isolated bacteria was detected by culture in appropriate media and checked for the zone of clearance. For the amylase assay, the bacteria were inoculated into amylase agar plates (containing 5.0 g of soluble starch, 5.0 g of peptone, 3.0 g of agar, 2.5 g of beef extract, 2.5 g of NaCl, and 500 mL of double-distilled water [ddH$_2$O] [pH 7.0 to 7.2]). The bacteria were cultured in PY broth for 24 h at 28°C with shaking at $200 \times g$, and the bacterial density was adjusted to $10^8$ CFU/mL. A 2-$\mu$L bacterial suspension was inoculated in amylase agar plates and cultured in an anaerobic workstation for 24 h at 28°C. Lugo's iodine solution was then added, and whether there was a clear circle was examined.

**Hemolysis and hydrophobicity assays.** To determine the hemolytic activity, 2 $\mu$L of bacterial suspension was inoculated on sheep blood agar plates (Huankai Biotechnology Co. Ltd., Guangdong, China) and cultured in an anaerobic workstation for 48 h at 28°C. Plates containing sheep blood agar were used as blank controls. The plates were observed for the generation of hemolytic zones, with no clearing (gamma-hemolysis), clean (beta-hemolysis), or greenish color (alpha-hemolysis) zone around the bacterial colonies.

To detect the hydrophobicity of the bacterium, the carbon hydroxyl bonding method was used in this experiment. According to the hydrophobicity of bacterial cell surface, H% = [(OD$_0$ − OD)/OD$_0$] × 100, where OD$_0$ represents the OD value of the bacterial suspension and OD represents the OD value of the bacterial suspension mixed with xylene. A volume of 2 mL of bacterial suspension was centrifuged at $5,000 \times g$ for 5 min, and the precipitate of the strain was collected. The precipitation was washed twice with 50 mM K$_2$HPO$_4$ buffer solution, the bacteria were resuspended with an appropriate amount of buffer solution, the concentration of bacterial suspension was adjusted (OD$_{560}$ = 1.0), and the buffer solution was used as the control. One milliliter of bacterial suspension and 4 mL of xylene were mixed fully and stabilized at 37°C for 1 h, and the aqueous value (OD$_{560}$) was tested.

***In vitro* fermentation experiment.** The candidate probiotic bacterium (*C. butyricum* G13) inoculated into PY broth at 28°C for 24 h with shaking at $200 \times g$ was used to prepare a bacterial suspension with $10^7$ CFU/mL. The gut contents were collected from 12 healthy mud crabs (body weight, appropriately 40 g). A volume of 100 $\mu$L of gut contents was added to the PY broth (5 mL) supplemented with sterile saline (control) and *C. butyricum* G13 (at $10^7$ CFU/mL) alone (G13) or in combination with GOS (0.5%) (G13+GOS), RS (0.5%) (G13+RS), or both GOS (0.5%) and RS (0.5%) (G13+GOS+RS). The culture tubes were incubated for 24 h at 28°C and shaken at $200 \times g$. The samples (2 mL) from the culture tubes were collected and centrifuged at $12,000 \times g$ for 10 min. The supernatants were used for the determination of SCFA production by gas chromatography, and the bacterial pellets were used for genomic DNA extraction. The genomic DNA was used as the template for PCR to amplify the V3-V4 region of the 16S rRNA gene. The sequencing was performed by a commercial company using an Illumina NovaSeq 6000 (Illumina Novogene, Tianjing, China).

**Bioinformatics and data analysis.** All obtained sequences were sorted based on their corresponding unique barcodes. The sequences of each sample were merged using Flash-1.2.8 (57). Sequences were multiplexed using a microbiome analysis program (QIIME1.8.0) and subjected to quality filtering (58). Chimeric sequences were detected through the comparison of high-quality sequences to the species annotation database and removed using the UCHIME algorithm (59). In order to study the species composition of each group, OTU cluster analysis was conducted with 97% identity, and species annotation was made based on the OTU sequences. The alpha diversity indices (community richness, Chao1 and ACE; evenness,

Shannon and Simpson), Good coverage, and sparsity curves were analyzed and calculated using the QIIME software package. Principal-coordinate analysis (PCoA) was performed using the UniFrac tool (60). Using the Galaxy/Hutlab (https://huttenhower.sph.harvard.edu/galaxy/) effect size (LEfSe), linear discriminant analysis was performed to determine significant differences in abundances between groups of microorganisms. PICRUSt1.0.0 was used to predict the functional spectrum of microbial communities (61). Cluster analysis was performed using PAST2.16. Differences obtained by the tests were considered statistically significant at a $P$ value of $<0.05$.

***Vibrio parahaemolyticus* infection test.** Commercial pellet feed was used as a basal feed (45% crude protein, 9% ether extract, 14% ash, and 12% moisture) (Guangdong Yuequn Marine Biological Research and Development Co., Ltd., China). *C. butyricum* G13 and *V. parahaemolyticus* were cultured in PY and LB medium, respectively, for 24 h. The bacterial culture was centrifuged, washed three times with 0.9% sterile saline, and suspended in sterile saline. The bacterial suspension (of *C. butyricum* G13 or *V. parahaemolyticus*) was sprayed in the basal feed. The experiment included four groups: the control (sterile saline) group and treated groups (*V. parahaemolyticus* [$1 \times 10^7$ mL/g] alone or in combination with *C. butyricum* G13 [$10^7$ CFU/g] [G13+*V. parahaemolyticus*] or *C. butyricum* G13 [$10^7$ CFU/g], GOS [0.5%], and RS [0.5%] [G13+GOS+RS+*V. parahaemolyticus*]). All feed was dried at 40°C before use. The bacterial concentrations used in this study were based on previous studies (10, 25, 62).

A total of 120 mud crabs (body weight, appropriately 30.4 g) purchased from a culture farm in Niutianyang (Shantou, Guangdong, China) were randomly divided into four groups. Mud crabs were tamed to the laboratory conditions (temperature, 27°C; pH 7.95; and salinity, 15‰). The mud crabs were fed a basal diet once a day (17:00) for 7 days. On day 8, each mud crab in the treated groups was orally administered 200 $\mu$L of *V. parahaemolyticus* (at $1 \times 10^7$ CFU/mL). The control group was treated with the same volume of 0.9% sterile saline. On days 8 and 9, mud crabs in each of the groups *V. parahaemolyticus*, G13, and G13+GOS+RS were fed with corresponding diets containing $1 \times 10^7$ CFU/g of *V. parahaemolyticus* to maintain the presence of pathogens in the gut. From day 10 onwards, the mud crabs in each group were fed with corresponding diets once daily (17:00). During the experimental period, feed residues and feces were removed and water was exchanged daily. The survival rates of mud crabs were recorded for 7 days.

After the challenge test, the mud crabs were disinfected and dissected on ice. Hepatopancreas and intestinal contents were quickly collected and stored in liquid nitrogen and then stored at $-80$°C. The intestines were collected and completely soaked in 4% tissue fixation solution for tissue section and H&E staining observation. The hepatopancreas was accurately weighed and mixed evenly according to the hepatopancreas (grams)/0.9% sterile saline ratio of 1:9. The supernatant was centrifuged at 2,500 $\times$ $g$ for 10 min under the condition of the ice water bath, and then 10% homogenate was obtained from the supernatant, which was used to detect enzyme activity, mainly including alkaline phosphatase (AKP), acid phosphatase (ACP), catalase (CAT), malonaldehyde (MDA), and superoxide dismutase (SOD), using the corresponding commercial test kits (Nanjing Jiengcheng Bioengineering Institute, China). Intestinal contents were used to detect changes in SCFA content after the challenge test. All animal handling procedures were reviewed and approved by the ethics committee of the "Regulations for the Administration of Affairs Concerning Experimental Animals."

**Data availability.** All data sets are available at the NCBI SRA database under BioProject number PRJNA948684.

## ACKNOWLEDGMENTS

This work was supported by grants from Li Ka Shing Foundation Cross-Disciplinary Research Grant (no. 2020LKSFG01E), National Natural Science Foundation of China (42076125, 41876152, and 31850410487), STU Scientific Research Foundation for Talents (NTF21027), Special Projects in Key Fields of General Universities in Guangdong Province (2022ZDZX4007 and 2021ZDZX4018), Guangdong provincial project of Science and Technology (2017B020204003), and Guangdong provincial Special Fund for Modern Agriculture Industry Technology Innovation Teams (2019KJ141).

We have no conflicts of interest to declare.

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
