## [Reviewer comments · Microbiology Spectrum]

Microbiology Spectrum

Identification and Characterization of Potential Probiotic, *Clostridium butyricum* G13, Isolated from the Intestine Mud Crab (*Scylla paramamosain*)

Huifen Liang, Ngoc Tuan Tran, Tiaoqiu Deng, Jinkun Li, Yifan Lei, Md. Akibul Hasan Bakky, Ming Zhang, Rui Li, Wenxuan Chen, Yueling Zhang, Xiuli Chen, and Shengkang Li

Corresponding Author(s): Ngoc Tuan Tran, Shantou University

Review Timeline:

Submission Date:	March 30, 2023
Editorial Decision:	April 22, 2023
Revision Received:	May 5, 2023
Accepted:	June 26, 2023

Editor: Konstantinos Kormas

Reviewer(s): Disclosure of reviewer identity is with reference to reviewer comments included in decision letter(s). The following individuals involved in review of your submission have agreed to reveal their identity: Huan Wang (Reviewer #1)

Transaction Report:

DOI: <https://doi.org/10.1128/spectrum.01317-23>

April 22, 2023

Dr. Ngoc Tuan Tran
Shantou University
No. 243, Daxue Road, Shantou City, Guangdong Province, China
Shantou
China

Re: Spectrum01317-23 (**Identification and Characterization of Potential Probiotic, *Clostridium butyricum* G13, Isolated from the Intestine Mud Crab (*Scylla paramamosain*)**)

Dear Dr. Ngoc Tuan Tran:

Link Not Available

Sincerely,

Konstantinos Kormas

Journals Department
Reviewer comments:

Reviewer #1 (Comments for the Author):

Recommendations- Spectrum01317-23

The manuscript entitled "Identification and Characterization of Potential Probiotic, *Clostridium butyricum* G13, Isolated from the Intestine Mud Crab (*Scylla paramamosain*)". The current manuscript provides some interesting points about the isolation of the butyrate-producing bacterium, *Clostridium butyricum* G13 from the intestine of mud crab. The study conducted the investigation of probiotic properties of the isolated bacterium and the results confirmed the safety and viability of the bacterium under the intestinal conditions, in vitro. The results of in vitro fermentation indicated the stimulation of producing short-chain fatty acids (especially butyrate) and modification of gut microbiota after 24 h post incubation. The authors also studied the effectiveness of

the bacterium or/and its combination with prebiotics in the protection of mud crabs against *Vibrio parahaemolyticus* infection. The study is technically well-conducted and presented. However, some points need to be clarified and revised before publication.

1. What are the limitations and strengths of the study? I suggest that the authors could supply new directions for further investigations from the findings found in this study in the Conclusion section.
2. Why the butyrate-producing bacteria rather than other bacterial groups were focused on in the present study? What is the effectiveness of this bacterial group in aquatic animals? Could we directly use the butyric acid or its salts as feed additives added to feeds for aquatic animals, why?
3. Colonization is one of the most important criteria for the selection of probiotic bacteria; however, in the current study, I found that the bacterium *Clostridium butyricum* G13 was isolated from the intestinal tract of mud crab, did the authors focus on the colonization of this bacteria in their study?
4. The reviewer found that the authors used the bacterial suspension of *Clostridium butyricum* G13 was 107 CFU/g feed or 107 CFU/mL in the in vivo challenge test or in the in vitro fermentation, respectively, please explain carefully the bacterial densities used and show the information in the manuscript.
5. Could the authors explain why *Vibrio parahaemolyticus*, instead of other pathogens, was used to challenge the mud crabs? and why the cell density of *Vibrio parahaemolyticus* (1×10^7 mL/g) was used to conduct the challenge test?
6. In the in vitro fermentation study, the relative abundance of the genus *Edwardsiella* was increased in the experimental groups added to *Clostridium butyricum* G13 or its combination with prebiotic(s). As we know that the members of *Edwardsiella* are the main pathogens causing diseases in freshwater fish, especially catfish (such as *Pangasianodon hypophthalmus*, *Ictalurus punctatus*), could the authors explain the obtained results herein?
7. The results of the study showed that *Clostridium butyricum* G13 can protect the host against *Vibrio parahaemolyticus* infection, so could the author explain the mechanism of action that *C. butyricum* G13 can prevent the infection?

Staff Comments:

Preparing Revision Guidelines

Please return the manuscript within 60 days; if you cannot complete the modification within this time period, please contact me. If you do not wish to modify the manuscript and prefer to submit it to another journal, please notify me of your decision immediately so that the manuscript may be formally withdrawn from consideration by Microbiology Spectrum.

Recommendations- Spectrum01317-23

The manuscript entitled “Identification and Characterization of Potential Probiotic, *Clostridium butyricum* G13, Isolated from the Intestine Mud Crab (*Scylla paramamosain*)”. The current manuscript provides some interesting points about the isolation of the butyrate-producing bacterium, *Clostridium butyricum* G13 from the intestine of mud crab. The study conducted the investigation of probiotic properties of the isolated bacterium and the results confirmed the safety and viability of the bacterium under the intestinal conditions, *in vitro*. The results of *in vitro* fermentation indicated the stimulation of producing short-chain fatty acids (especially butyrate) and modification of gut microbiota after 24 h post incubation. The authors also studied the effectiveness of the bacterium or/and its combination with prebiotics in the protection of mud crabs against *Vibrio parahaemolyticus* infection. The study is technically well-conducted and presented. However, some points need to be clarified and revised before publication.

1. What are the limitations and strengths of the study? I suggest that the authors could supply new directions for further investigations from the findings found in this study in the Conclusion section.
2. Why the butyrate-producing bacteria rather than other bacterial groups were focused on in the present study? What is the effectiveness of this bacterial group in aquatic animals? Could we directly use the butyric acid or its salts as feed additives added to feeds for aquatic animals, why?
3. Colonization is one of the most important criteria for the selection of probiotic bacteria; however, in the current study, I found that the bacterium *Clostridium butyricum* G13 was isolated from the intestinal tract of mud crab, did the authors focus on the colonization of this bacteria in their study?
4. The reviewer found that the authors used the bacterial suspension of *Clostridium butyricum* G13 was 10^7 CFU/g feed or 10^7 CFU/mL in the *in vivo* challenge test or in the *in vitro* fermentation, respectively, please explain carefully the bacterial densities used and show the information in the manuscript.
5. Could the authors explain why *Vibrio parahaemolyticus*, instead of other pathogens, was used to challenge the mud crabs? and why the cell density of *Vibrio parahaemolyticus* (1×10^7 mL/g) was used to conduct the challenge test?
6. In the *in vitro* fermentation study, the relative abundance of the genus *Edwardsiella* was increased in the experimental groups added to *Clostridium butyricum* G13 or its combination with prebiotic(s). As we know that the members of *Edwardsiella* are the main pathogens causing diseases in freshwater fish, especially catfish (such as *Pangasianodon hypophthalmus*, *Ictalurus punctatus*), could the authors explain the obtained results herein?

7. The results of the study showed that *Clostridium butyricum* G13 can protect the host against *Vibrio parahaemolyticus* infection, so could the author explain the mechanism of action that *C. butyricum* G13 can prevent the infection?

Response to Reviewers

Dear editor,

We greatly appreciated the editor and the anonymous reviewers for critical reading of the manuscript and giving so many helpful and constructive advices. The manuscript has been carefully revised and greatly improved for related information according to the comments and suggestions of the editor and the reviewers. We indicated the changes in the revised manuscript to highlight in yellow.

The point-to-point answers to the comments of the reviewers are as follows:

To Editors:

We have added two authors: Yueling Zhang and Xiuli Chen to our manuscript. In our submitted manuscript, due to some objective reasons, we forgot the contribution of these authors. Actually, they helped us in guiding our experimental designs and reviewing the manuscript. Therefore, in this revised manuscript, these authors have been added. Thanks for your understanding.

To Reviewer:

The manuscript entitled "Identification and Characterization of Potential Probiotic, Clostridium butyricum G13, Isolated from the Intestine Mud Crab (Scylla paramamosain)". The current manuscript provides some interesting points about the isolation of the butyrate-producing bacterium, Clostridium butyricum G13 from the intestine of mud crab. The study conducted the investigation of probiotic properties of the isolated bacterium and the results confirmed the safety and viability of the bacterium under the intestinal conditions, in vitro. The results of in vitro fermentation indicated the stimulation of producing short-chain fatty acids (especially butyrate) and modification of gut microbiota after 24 h post incubation. The authors also studied the effectiveness of the bacterium or/and its combination with prebiotics in the protection of mud crabs against Vibrio parahaemolyticus infection. The study is technically well-conducted and presented. However, some points need to be clarified and revised before publication.

Reply: Many thanks for your comments.

1. What are the limitations and strengths of the study? I suggest that the authors could

supply new directions for further investigations from the findings found in this study in the Conclusion section.

Reply: Thank you for your suggestions. The advantage of this study is that it was based on the results of our previous study (1). In our previous study, the *in vitro* fermentation method using gut contents of mud crabs was conducted and demonstrated to be a suitable strategy for the isolation of butyrate-producing bacteria. In this current study, the butyric-producing bacterium, *Clostridium butyricum* G13, was the first time isolated from mud crab (*Scylla paramamosain*) using *in vitro* fermentation. The probiotic potential of *C. butyricum* G13 was demonstrated profoundly through the *in vitro* characterization and the protective role of the bacterium in mud crab against *Vibrio parahaemolyticus* infection was also proven through *in vivo* challenge test. However, the limitation is that the effects of dietary supplementation of *C. butyricum* G13 on growth, physiological changes, and gut microbiota of mud crabs have not been conducted yet. Also, the molecular mechanisms that *C. butyricum* G13 protects mud crabs against *V. parahaemolyticus* infection have not been elucidated. The new directions for further investigations from our findings were supplied in the Conclusion section in the revised manuscript (Lines: 354-357).

2. Why the butyrate-producing bacteria rather than other bacterial groups were focused on in the present study? What is the effectiveness of this bacterial group in aquatic animals? Could we directly use the butyric acid or its salts as feed additives added to feeds for aquatic animals, why?

Reply: Previously, it has been known that *Clostridium butyricum* is more resistant to low pH, high-temperature environments, and multiple antibiotics than other probiotics such as *Bacillus*, *Lactobacillus*, and yeast (2). The results of our study also found that *C. butyricum* G13 grew well in a wide range of pH (4-9), NaCl (1-2.5%), and bile salt concentration (0.2-1.0%), indicating its advantage when growing in the host's intestinal environment. *Clostridium butyricum* G13 can use carbohydrates and produce large amounts of short-chain fatty acids, especially butyric acid. Butyric acid has been proven beneficial for the regeneration and repair of the intestinal epithelium and regulation of intestinal health microecology (3). To our knowledge, *C. butyricum* can directly by itself or/and indirectly by its metabolites (especially butyric acid) improve the health of their

hosts, which is the advantage of the use of *C. butyricum* as probiotics. In our previous studies, *C. butyricum* has been reviewed to have significant probiotic benefits in improving growth, feed utilization, immune response, pathogenic resistance, and stress reduction in aquatic animals (2, 3).

“Could we directly use the butyric acid or its salts as feed additives added to feeds for aquatic animals, why?”, we think that it is “YES”, but the issues regarding the use of butyrate, including the types and optimal dose, are required to be focused in further studies. Although numerous reports have demonstrated the efficiency of butyrate or its salts in aquatic animals, the previous findings showed the effects of pure form and high levels of butyrate may cause fat accumulation in the liver and the inflammation and destruction of cells (4). Also, butyric acid is a volatile fatty acid, we think that under laboratory conditions the manipulation of adding butyrate or its salts to feed can be performed. However, under farming conditions, it is carefully considered when the amount of feed is too large and takes a long time from preparation to feeding. Taken together, we suggest that the application of butyrate-producing bacteria is more suitable for enhancing the health status of aquatic animals. For example, the previous study indicated that *C. butyricum* is more beneficial than the use of butyrate (and its forms) (3). The daily application of butyrate can increase production costs, meanwhile, *C. butyricum* can colonize the gut of its host for a long time, and thus it is not necessary to supplement daily. Application of *C. butyricum* can produce butyric acid and promote the growth of other butyrate-producing microbes, subsequently conferring health benefits on the host.

3. Colonization is one of the most important criteria for the selection of probiotic bacteria; however, in the current study, I found that the bacterium *Clostridium butyricum* G13 was isolated from the intestinal tract of mud crab, did the authors focus on the colonization of this bacteria in their study?

Reply: Yes, we agree that colonization is an important factor for a potential probiotic bacterium. In this study, the hydrophobicity assay was designed to test the hydrophobicity of the bacteria, and the contact enzyme activity was tested. The results proved that the hydrophobicity of *C. butyricum* G13 was 43.8% (OD560 of suspensions containing xylene and without xylene were 1.345 and 0.907, respectively), indicating that the bacterium has

strong adhesion ability to the intestinal tract. However, the *in vivo* evaluation of the colonization of *C. butyricum* G13 has not been conducted yet, which is also one of the limitations of our present study.

4. The reviewer found that the authors used the bacterial suspension of *Clostridium butyricum* G13 was 10^7 CFU/g feed or 10^7 CFU/mL in the *in vivo* challenge test or in the *in vitro* fermentation, respectively, please explain carefully the bacterial densities used and show the information in the manuscript.

Reply: In this study, the concentration of *C. butyricum* G13 (10^7 CFU/g feed or 10^7 CFU/mL) used in this study was followed by the method previously reported by Li (5). The information has been added to the revised manuscript (Lines 508- 509).

5. Could the authors explain why *Vibrio parahaemolyticus*, instead of other pathogens, was used to challenge the mud crabs? and why the cell density of *Vibrio parahaemolyticus* (1×10^7 mL/g) was used to conduct the challenge test?

Reply: Thanks for your comments. Mud crab is one of the most important commercial mariculture species in southern China. Under culture conditions, the mud crabs are usually found to be infected by the bacterium *Vibrio parahaemolyticus*. In fact, *V. parahaemolyticus* has been known as mainly responsible for disease outbreaks and caused large economic losses in mud crab production in China (6-8). For these reasons, *V. parahaemolyticus* was used as a pathogen for evaluating the effectiveness of the probiotic potential of *C. butyricum* G13. In this current study, the cell density of *V. parahaemolyticus* (1×10^7 mL/g) used was based on the previous report of Zhang (6). The information has been added to the revised manuscript (Lines: 508- 509).

6. In the *in vitro* fermentation study, the relative abundance of the genus *Edwardsiella* was increased in the experimental groups added to *Clostridium butyricum* G13 or its combination with prebiotic(s). As we know that the members of *Edwardsiella* are the main pathogens causing diseases in freshwater fish, especially catfish (such as *Pangasianodon hypophthalmus*, *Ictalurus punctatus*), could the authors explain the obtained results herein?

Reply: Thanks for your comments. In the *in vitro* fermentation study, the results found that the relative abundance of the genus *Edwardsiella* was increased in the

experimental groups added to *C. butyricum* G13 singly or its combination with prebiotics. As shown in the Discussion, we discussed on this “Previous studies have demonstrated that the members of the genus *Edwardsiella* are generally considered pathogens of aquatic animals (9, 10). However, in a recent study, the heat-inactivated *Edwardsiella* sp. 34HN has been shown a probiotic bacterium, which could improve growth performance and enhance the immune response of African catfish (*Clarias gariepinus*) during 100 days of feeding” (11) (Lines: 314-317). Therefore, we speculate that the members of the genus *Edwardsiella* are pathogens to this host but not necessarily harmful to others. Moreover, mud crabs live in marine water, so it is possible that the pathogenic species (belonging to the genus *Edwardsiella*) are absent in this environment. Interestingly, the two bacterial species, *Edwardsiella ictaluri* and *Edwardsiella tarda* have been known as pathogens in catfish (including *Pangasianodon hypophthalmus* and *Ictalurus punctatus*) (12, 13). However, these species have not yet been reported in mud crabs, which merits further investigation.

7. The results of the study showed that *Clostridium butyricum* G13 can protect the host against *Vibrio parahaemolyticus* infection, so could the author explain the mechanism of action that *C. butyricum* G13 can prevent the infection?

Reply: In the *in vitro* study, we found that *C. butyricum* G13 can inhibit the growth of many pathogens, including *Vibrio parahaemolyticus*, *Staphylococcus aureus*, *Vibrio harveyi*, *Vibrio alginolyticus*, and *Aeromonas hydrophila*. In the challenge test, the results indicated that *C. butyricum* G13 alone or in combination with prebiotics can enhance the survival rate of mud crabs infected with *V. parahaemolyticus*. However, the mechanism of action that *C. butyricum* G13 can prevent the infection has not been known yet. Combined with other findings we suggest that *C. butyricum* G13 produce butyric acid which can inhibit the growth of *V. parahaemolyticus*. Also, *C. butyricum* G13 and its derived butyric acid may modulate the gut microbiota in the way beneficial for hosts and simultaneously inhibit the growth of *V. parahaemolyticus*. Moreover, *C. butyricum* and its derived butyric acid have been reported to have beneficial effects on the growth, proliferation, differentiation and mucosal formation of intestinal epithelial cells, and can improve the immune capacity of the host and has anti-inflammatory effects (14). Collectively, the mechanism of action that *C. butyricum* G13 affecting the growth of *V. parahaemolyticus*

merits further investigation.

Reference

1. Tran NT, Tang Y, Li ZZ, Zhang M, Wen XB, Ma HY, Li SK. 2020. Galactooligosaccharides and resistant starch altered microbiota and short-chain fatty acids in an *in vitro* fermentation study using gut contents of mud crab (*Scylla paramamosain*). *Frontiers in Microbiology* 11:1352. <https://doi.org/10.3389/fmicb.2020.013522>.
2. Tran NT, Li ZZ, Ma HY, Zhang YL, Zheng HP, Gong Y, Li SK. 2020. *Clostridium butyricum*: a promising probiotic confers positive health benefits in aquatic animals. *Reviews in Aquaculture* 12(4):2573–2589. <https://doi.org/10.1111/raq.12459>.
3. Tran NT, Liang HF, Li JK, Deng TQ, Zhang M, Li SK. 2023. Health benefits of butyrate and its producing bacterium, *Clostridium butyricum*, on aquatic animals. *Fish & Shellfish Immunology Reports* 4:100088. <https://doi.org/10.1016/j.fsirep.2023.100088>.
4. Jesus GF, Pereira SA, Owatari MS, Syracuse N, Silva BC, Silva A, Pierri BS, Lehmann NB, Figueiredo HC, Fracalossi DM. 2019. Protected forms of sodium butyrate improve the growth and health of Nile tilapia fingerlings during sexual reversion. *Aquaculture* 499:119-127. <https://doi.org/10.1016/j.aquaculture.2018.09.027>.
5. Li ZZ. 2018. Identification of prebiotics and butyrate-producing probiotics, and their applications on *Nibea coibor*. Shantou University, Ph.D. Dissertation [Chinese].
6. Zhang XS, Tang XX, Tran NT, Huang Y, Gong Y, Zhang YL, Zheng HP, Ma HY, Li SK. 2019. Innate immune responses and metabolic alterations of mud crab (*Scylla paramamosain*) in response to *Vibrio parahaemolyticus* infection. *Fish & Shellfish Immunology* 87:166–177. <https://doi.org/10.1016/j.fsi.2019.01.011>.
7. Ye HH, Tao Y, Wang GZ, Lin QW, Chen XL, Li SJ. 2011. Experimental nursery culture of the mud crab *Scylla paramamosain* (Estampador) in China. *Aquaculture* 19:313–321. <https://doi.org/s10499-010-9399-3>.
8. Wu HJ, Sun LB, Li CB, Li ZZ, Zhang Z, Wen XB, Li SK. 2014. Enhancement of the immune response and protection against *Vibrio parahaemolyticus* by indigenous probiotic *Bacillus* strains in mud crab (*Scylla paramamosain*). *Fish & Shellfish Immunology* 41:156–162. <https://doi.org/10.1016/j.fsi.2014.08.027>.

9. Janda JM, Abbott SL, Kroske-Bystrom S, Cheung WK, Powers C, Kokka RP, Tamura K. 1991. Pathogenic properties of *Edwardsiella* species. *Journal of Clinical Microbiology* 29(9):1997–2001. <https://doi.org/10.1128/jcm.29.9.1997-2001.1991>.
10. Pirarat N, Kobayashi T, Katagiri T, Maita M, Endo M. 2006. Protective effects and mechanisms of a probiotic bacterium *Lactobacillus rhamnosus* against experimental *Edwardsiella tarda* infection in tilapia (*Oreochromis niloticus*). *Veterinary Immunology and Immunopathology* 113(3-4):339–347. <https://doi.org/10.1016/j.vetimm.2006.06.003>.
11. Selim KM, El-Sayed HM, El-Hady MA, Reda RM. 2019. *In vitro* evaluation of the probiotic candidates isolated from the gut of *Clarias gariepinus* with special reference to the *in vivo* assessment of live and heat-inactivated *Leuconostoc mesenteroides* and *Edwardsiella* sp. *Aquaculture International* 27:33–51. <https://doi.org/10.1007/s10499-018-0297-4>.
12. Meyer FP, Bullock GL. 1973. *Edwardsiella tarda*, a new pathogen of channel catfish (*Ictalurus punctatus*). *Applied Microbiology* 25(1):155–156. <https://doi.org/10.1128/am.25.1.155-156.1973>.
13. Bartie KL, Austin FW, Diab A, Dickson C, Dung TT., Giacomini M, Crumlish M. 2012. Intraspecific diversity of *Edwardsiella ictaluri* isolates from diseased freshwater catfish, *Pangasianodon hypophthalmus* (Sauvage), cultured in the Mekong Delta, Vietnam. *Journal of Fish Diseases* 35(9):671–682. <https://doi.org/10.1111/j.1365-2761.2012.01376.x>.
14. Venegas DP, De la Fuente MK, Landskron G, González MJ, Quera R, Dijkstra G, Harmsen HJM, Faber KN, Hermoso MA. 2019. Short chain fatty acids (SCFAs)-mediated gut epithelial and immune regulation and its relevance for inflammatory bowel diseases. *Frontiers in Immunology* 10:277. <https://doi.org/10.3389/fimmu.2019.00277>.

Yours Sincerely,

Shengkang Li

Institute of Marine Sciences

Shantou University

June 26, 2023

Dr. Ngoc Tuan Tran
Shantou University
No. 243, Daxue Road, Shantou City, Guangdong Province, China
Shantou
China

Re: Spectrum01317-23R1 (**Identification and Characterization of Potential Probiotic, *Clostridium butyricum* G13, Isolated from the Intestine Mud Crab (*Scylla paramamosain*)**)

Dear Dr. Ngoc Tuan Tran:

Your manuscript has been accepted, and I am forwarding it to the ASM Journals Department for publication. You will be notified when your proofs are ready to be viewed.

Sincerely,

Konstantinos Kormas
Editor, Microbiology Spectrum
